# Simultaneous Imaging and Therapy Using Epitope-Specific Anti-Epidermal Growth Factor Receptor (EGFR) Antibody Conjugates

**DOI:** 10.3390/pharmaceutics14091917

**Published:** 2022-09-10

**Authors:** Anjong Florence Tikum, Anand Krishnan Nambisan, Jessica Pougoue Ketchemen, Hanan Babeker, Musharraf N. Khan, Emina E. Torlakovic, Humphrey Fonge

**Affiliations:** 1Department of Medical Imaging, College of Medicine, University of Saskatchewan, Saskatoon, SK S7N 0W8, Canada; 2Department of Pathology and Lab. Medicine, College of Medicine, University of Saskatchewan, 107 Wiggins Rd, Saskatoon, SK S7N 5A2, Canada; 3Department of Medical Imaging, Royal University Hospital Saskatoon, Saskatoon, SK S7N 0W8, Canada

**Keywords:** ^89^Zr-matuzumab, epitope-specific immunoconjugates, nimotuzumab antibody drug conjugate, PET/CT imaging, colorectal cancer EGFR

## Abstract

Matuzumab and nimotuzumab are anti-EGFR monoclonal antibodies that bind to different epitopes of domain III of EGFR. We developed ^89^Zr-matuzumab as a PET probe for diagnosis/monitoring of response to treatment of a noncompeting anti-EGFR nimotuzumab antibody drug conjugate (ADC) using mouse colorectal cancer (CRC) xenografts. We developed ^89^Zr-matuzumab and performed quality control in EGFR-positive DLD-1 cells. The K_D_ of matuzumab, DFO-matuzumab and ^89^Zr-matuzumab in DLD-1 cells was 5.9, 6.2 and 3 nM, respectively. A competitive radioligand binding assay showed that ^89^Zr-matuzumab and nimotuzumab bound to noncompeting epitopes of EGFR. MicroPET/CT imaging and biodistribution of ^89^Zr-matuzumab in mice bearing EGFR-positive xenografts (HT29, DLD-1 and MDA-MB-231) showed high uptake that was blocked with pre-dosing with matuzumab but not with the noncompeting binder nimotuzumab. We evaluated nimotuzumab-PEG_6_-DM1 ADC in CRC cells. IC_50_ of nimotuzumab-PEG_6_-DM1 in SNU-C2B, DLD-1 and SW620 cells was dependent on EGFR density and was up to five-fold lower than that of naked nimotuzumab. Mice bearing the SNU-C2B xenograft were treated using three 15 mg/kg doses of nimotuzumab-PEG_6_-DM1, and ^89^Zr-matuzumab microPET/CT was used to monitor the response to treatment. Treatment resulted in complete remission of the SNU-C2B tumor in 2/3 mice. Matuzumab and nimotuzumab are noncompeting and can be used simultaneously.

## 1. Introduction

Metastatic colorectal cancer (mCRC) is the second leading cause of death from cancer with a five-year survival rate of <10% (stage IV) [1], likely due to the fact that over 45% of CRC patients have metastatic disease at initial diagnosis. Surgery, which is a primary treatment option, is contraindicated in patients with advanced disease, and even when it is possible, the local recurrence rate is very high (38–88%) [2]. About 80% of CRC patients overexpress epidermal growth factor receptor (EGFR) [3,4]. Kirsten rat sarcoma viral oncogene (KRAS) is an intracellular effector molecule that routs ligand-bound EGFR to the nucleus, where it stimulates proliferation. Mutations in the KRAS oncogene (present in >40% of mCRC) lead to constitutive overactivation of EGFR and drives de novo resistance to anti-EGFR drugs [5,6,7]. EGFR is a transmembrane receptor tyrosine kinase belonging to the ErbB family, and is also expressed in different tissues, including lung, skin, hair follicles, and gastrointestinal tract [8]. EGFR signaling leads to cell growth, differentiation, proliferation, and inhibition of apoptosis [9,10].

Anti-EGFR antibodies such as cetuximab and panitumumab [11,12,13,14] have been approved for treating different EGFR-positive cancers. With the exception of nimotuzumab, the use of these antibodies is linked to significant cutaneous toxicity in 45–100% of patients [15,16,17]. On the other hand, nimotuzumab is well-tolerated [18,19] and has low skin toxicities because of its “affinity optimized” binding characteristic [20].

Though the clinical use of these unconjugated monoclonal antibodies (mAb) has increased over the years, most of them are used in combination with chemotherapy [21,22]. The efficacy of mAbs is increased by the conjugation with potent cytotoxic agents to generate antibody drug conjugates (ADC), and many of these have been approved or are in clinical development [23]. Maytansine (DM1) is the drug of choice for many ADCs in clinical development [24]. Despite improvements in efficacy with ADCs, acquired resistance is common, and this is mostly due to the expression of the multidrug-resistant gene (MDR1) [25]. Recently, it has been shown that acquired resistance due to MDR1 can be circumvented using pegylated techniques [26,27]. Furthermore, pegylated ADCs are more hydrophilic, making it possible to conjugate many drugs to the antibodies without adverse effects on the binding to antigens and pharmacokinetics [26,27].

Non-invasive molecular imaging using positron emission tomography (PET) offers many advantages over ex vivo methods that involve the use of biopsies such as immunohistochemistry (IHC) [28,29]. The simultaneous use of antibodies that bind to different epitopes/domains on the same receptor is advantageous because they can be used in combination to enhance therapeutic effects and/or one of them can be used as an imaging agent to provide accurate response monitoring to the other being used as a therapeutic. In this work, using mostly KRAS mutant and wild-type models of mCRC, we evaluated the imaging characteristics of ^89^Zr-matutuzmab using microPET/CT and the therapeutic efficacy of nimotuzumab-PEG_6_-DM1. We also evaluated the potential for the simultaneous use of ^89^Zr-matuzumab PET/CT and nimotuzumab-PEG_6_-DM1 for biparatopic imaging and therapy, respectively, in EGFR-positive KRAS mutant colorectal cancer models.

## 2. Materials and Methods

All reagents and solvents obtained from commercial suppliers were used without any further purification. DM1 was obtained from Toronto Research Chemical (Toronto, ON, Canada), and NHS-PEG_6_-maleimide was purchased from Biochempeg (Watertown, MA, USA). Colorectal cancer cell lines DLD-1 (RRID:CVCL_0248), SNU-C2B (RRID:CVCL_1710), HT-29 (RRID:CVCL_0320), and SW620 (RRID:CVCL_0547), and breast cancer cell line MDA-MB-231 (RRID:CVCL_0062) with different levels of EGFR expression were obtained from ATCC (Manassas, VA, USA). The cells were cultured in RPMI-1640 (DLD-1 and SNU-C2B), Myco’s 5A media (HT-29), and DMEM (SW620, MDA-MB-231). All cell lines were authenticated using short tandem repeat (STR) profiling at the Centre for Applied Genomics (Hospital for SickKids, Toronto, ON, Canada). The cells were free of mycoplasma prior to use. All cultured media used were supplemented with 10% FBS and 1% penicillin/streptomycin. Biosimilar anti-EGFR antibodies nimotuzumab and matuzumab were acquired from Ichorbio (Wayne, PA, USA) and were buffer-exchanged prior to use. The drug linker maytansine (DM1)-PEG_6_-NHS ester (DM1-PEG_6_-NHS) was synthesized and characterized as reported and was used to synthesize nimotuzumab-PEG_6_-DM1 ADC with a drug-to-antibody ratio (DAR) of 3–4 as reported previously by our group [30]. The DAR was determined using a UV spectrophotometric method reported earlier [31]. The analysis of molecular weight and purity of all the conjugated samples was performed on an Agilent 2100 Bioanalyzer system using an Agilent High Sensitivity Protein 250 Kit (cat # 5067-1575) according to the manufacturer’s protocol. Size-exclusion HPLC (SEC-HPLC) was performed using Waters 2796 Bioseparation modules, a Water 2487 l Absorbance Detector, and an XBridge^®^ BEH 200A SEC 3.5 μm 7.8 × 150 mm column (Waters Corporation, Milford, MA, USA) was used to determine the integrity of the conjugates and free drug or deferoxamine (DFO). The UV-Detector was set at 254 and 280 nm with PBS as the mobile phase and a flow rate of 0.45 mL/min.

### 2.1. Synthesis of Matuzumab Deferoxamine (DFO)

Matuzumab was conjugated with *p*-SCN-Bn-deferoxamine (DFO) for labeling with ^89^Zr as described previously [32]. Briefly, matuzumab (10 mg/mL in PBS) was buffer-exchanged in centrifugal filters (Amicon Ultra-4 Centrifugal Filter 30K NMCO, EMD Millipore, Burlington, MA, USA) using a 0.1 M NaHCO_3_ buffer (pH 9). Fifteen-fold molar excess of DFO was added to this mixture, and the reaction was allowed to react at 4 °C for 18 h. Excess unconjugated chelator was removed from the reaction mixture using a centrifuge and PBS as the storage buffer. Quality control of the immunoconjugate was done using SEC HPLC, flow cytometry, and a bioanalyzer.

### 2.2. Flow Cytometry of Immunoconjugates

Flow cytometry analysis was performed to determine the binding of the immunoconjugate to EGFR in comparison with that of the unconjugated antibodies. To do this, cells (DLD-1 or SNU-C2B) were collected, washed with PBS, and seeded in 96-well plates with each well containing 0.5 × 10^6^ cells. The cells were then treated with nimotuzumab, nimotuzumab-PEG_6_-DM1, matuzumab, or matuzumab-DFO (2000–0.011 nM) and incubated at 4 °C for 30 min. The cells were washed 3 times with cold PBS and then resuspended in PE-labeled goat anti-human IgG (1:100 dilution) secondary antibody (eBioscience^TM^, Burlington, ON, Canada) and incubated for 30 min at 4 °C. The cells were washed 3 times with PBS and resuspended in PBS. Flow data were acquired using Beckman Coulter Life Science cytoflex benchtop flow cytometer, and the results were analyzed using FlowJo v10 and GraphPad prism 9.

### 2.3. In Vitro Cytotoxicity of Nimotuzumab-PEG_6_-DM1

The in vitro cytotoxicity of nimotuzumab and nimotuzumab-PEG_6_-DM1 was determined using an IncuCyte S3 Live cell imaging system (Essen BioScience, Ann Arbor, MI, USA) in DLD-1, HT-29, SW620, and SNU-C2B cell lines. Briefly, cells were seeded in a 96-well flat bottom corning pre-coated with poly-D-lysine plates (10^4^ cells per well) 24 h before treatment. The following day, the cell media were removed, and the cells were washed with PBS. The cells were then treated with different concentrations (2 µM–0.033 nM) of nimotuzumab or nimotuzumab-PEG_6_-DM1 in growth media containing Incucyte^®^ Cytotox Red reagent and incubated at 37 °C for 30 min before imaging. Live-cell images were captured every 2 h using a 10× objective lens using phase contrast and a fluorescence channel. During each scan, 4 images were acquired until the end of the experiment. All cell images were processed and analyzed using Incucyte S3 software. The relative fluorescence values generated were used to calculate the IC_50_ values using GraphPad prism 9.

### 2.4. Radiolabeling with ^89^Zr and Radioligand Binding Assays

Radiolabeling and purification of DFO-matuzumab with ^89^Zr was done as reported in the literature [32]. Briefly, 0.1 M HEPES pH 7.4 was added to ^89^Zr-oxalate and kept at room temperature for 10 min. Then, 2 M Na_2_CO_3_ (pH 11) was added drop-wise while measuring the pH to neutralize the solution (pH 7 ± 0.2). DFO-matuzumab was then added to ^89^Zr at a specific activity of 0.5 MBq/µg, and the reaction mixture was incubated at 37 °C on a shaker at 700 RPM for 90 min. Saturated radioligand binding assay of ^89^Zr-matuzumab was done using colorectal cancer DLD-1 and breast cancer MDA-MB-231 cells. The cells were harvested and resuspended in cold PBS (0.5 × 10^6^ cells per tube). The cells were centrifuged at 1200 rpm for 5 min, and PBS was removed. For total binding, cells were incubated with the radioimmunoconjugate (0.2–95 nmol/L in 100 μL PBS) for 4 h at 4 °C. Nonspecific binding (NSB) was determined in a similar assay but in the presence of a 50-fold molar excess of unlabeled matuzumab (relative to the highest concentration of ^89^Zr-matuzumab). The cells were then washed with PBS buffer, and the activity in the cells was measured using a gamma counter (Wallac Wizard 1480, PerkinElmer, Waltham, MA, USA). Specific binding was obtained by subtracting total binding from nonspecific binding, and a nonlinear regression analysis with one-site binding equation was used to determine K_D_ using GraphPad Prism 9.

It is documented that both matuzumab and nimotuzumab bind to domain III of EGFR [33]. However, it is unknown whether they bind to different epitopes of domain III. To investigate this, a competition binding assay was performed. Briefly, DLD-1 cells (0.5 × 10^6^) were transferred into 3 vials incubated with ^89^Zr-matuzumab, but with slight modifications. The first vial was incubated with ^89^Zr-matuzumab only, the second vial was initially incubated with cold matuzumab whose concentration was 50× the concentration of radiolabeled ^89^Zr-matuzumab for blocking (nonspecific binding). After 2 h of pre-incubation with unlabeled matuzumab, ^89^Zr-matuzumab was added to these cells. The third vial was incubated with unlabeled nimotuzumab whose concentration was 50× the concentration of radiolabeled ^89^Zr-matuzumab. After 2 h of pre-incubation with nimotuzumab, ^89^Zr-matuzumab was added to these cells. After incubation, the cells were centrifuged at 1200 rpm, and the supernatant was collected separately. The experiment was performed in triplicates using three different concentrations of 150 nM and 75 nM ^89^Zr-matuzumab.

### 2.5. Tumor Xenografts, Micropet/CT Imaging, Biodistribution, and Pharmacokinetics

All animal studies were approved by the University of Saskatchewan Animal Care and Use Committee following the guidelines outlined in the Use of Laboratory Animals (protocol # 20170084). Female CD-1 nude mice were obtained from Charles River Canada (St-Constant, Quebec, QC, Canada) at 4 weeks of age and housed in a 12 h light, 12 h dark cycle in a temperature- and humidity-controlled vivarium. The animals had ad libitum access to food (Lab Diet, St. Louis, Missouri, MO, USA) and water [34]. After one week of acclimatization, the mice were subcutaneously injected with a suspension of EGFR-positive DLD-1 (5 × 10^6^), HT-29 (5 × 10^6^), SNU-C2B (10 × 10^6^), and SW620 (10 × 10^6^) colorectal cancer or MDA-MB-231 breast cancer cells (10 × 10^6^) in 100 μL of a 1:1 mixture of serum-free MEM/EBSS medium (HyClone Laboratories, Logan, UT, USA) and matrigel matrix basement membrane (Discovery Laboware Inc., Bedford, MA, USA) at the hind limb of each mouse. Xenografts were located in the right or left thigh of the hind legs of mice, and their tumor growth was followed with caliper measurements.

Mice bearing EGFR-positive colorectal cancer DLD-1, HT29, or SW620 (*n* ≥ 3/group) and breast cancer xenografts MDA-MB-231 were injected via a tail vein with 10–12 MBq of ^89^Zr-matuzumab (specific activity of 0.5 MBq/µg) followed by µPET/CT imaging at 24, 48, 72, and 120 h post-injection using a Vector4CT scanner (MILabs, Utrecht). The images were analyzed using PMOD 3.7 biomedical image analyzing software (PMOD, Davos, Switzerland). Additional groups of mice were sacrificed at 24 and 120 h post-injection, and activity in different organs was measured using a gamma counter (Wallac Wizard 1480, PerkinElmer, Waltham, MA, USA) and expressed as the percentage of injected activity per gram (% IA/g).

Pharmacokinetics of ^89^Zr-matuzumab were determined in healthy athymic nude mice (*n* = 3). The mice were injected with 3 ± 0.14 MBq of ^89^Zr-matuzumab via a tail vein, and blood was collected at different time points from a vein in heparinized capillary tubes. Activity in the capillary tube was measured using a gamma counter and expressed as % IA/mL. All relevant pharmacokinetic parameters were determined using an exponential decay curve fitting from a sigma plot using GraphPad Prism 9.

### 2.6. Nimotuzumab ADC Immunotherapy

When the tumor xenograft had reached an average size of 200 ± 100 mm^3^, the mice were randomized into 3 different groups (3–4 mice/group), and each mouse was injected intravenously with saline or 15 mg/kg of nimotuzumab or nimotuzumab-PEG_6_-DM1 that bound to a different epitope of EGFR from the imaging agent ^89^Zr-matuzumab on days 0, 6, and 11. To study if ^89^Zr-matuzumab could be used to evaluate the response to nimotuzumab/nimotuzumab ADC therapy, these tumor-bearing mice were injected with ^89^Zr-matuzumab followed by nimotuzumab/nimotuzumab ADC therapy (baseline imaging, day 0, and on day 20 after initialization of therapy). MicroPET/CT images were acquired at 24 h post-^89^Zr-matuzumab injection using a Vector4CT scanner (MILabs, Utrecht, The Netherlands) as described above.

### 2.7. Histology and Immunohistochemistry

Mice in treatment studies were sacrificed on day 51, and tumors were excised for histology. The tumors were fixed in 10% formalin and embedded in paraffin wax. Seven-micron sections were stained using Hematoxylin and Eosin (H&E) as per standard protocol. The slides were scanned using an Aperio ScanScope XT with a 200× objective lens. Immunohistochemistry (IHC) using mouse anti-Ki-67 (Clone MIB1; Dako Canada Inc., Mississauga, Ontario, Canada Immunostar Inc.; 1 in 50 dilution) was conducted at Prairie Diagnostic Services Inc., Saskatoon, Saskatchewan, using an automated slide stainer (Autostainer Plus, Dako Canada Inc., Mississauga, Ontario, ON, Canada). Terminal deoxynucleotidyl Transferase-Mediated dUTP-biotin Nick End-labeling (TUNEL) assay was performed using a commercial apoptosis detection kit (ApopTag^®^ Peroxidase In Situ Apoptosis Detection Kit from Millipore) following the manufacturer’s protocol. Images were acquired using an Aperio ScanScope XT with a 200× objective lens. For both Ki-67 IHC and TUNEL assays, the percent of positive cells was determined in areas away from and in the band of viable tumor tissue surrounding necrosis. Apoptotic bodies were also counted on H&E sections.

### 2.8. Statistical Analysis

Unless otherwise stated, all data were expressed as the mean ± standard deviation (SD) or standard error of mean (SEM) of at least 3 independent experiments. Statistical comparisons between the experimental groups were performed using two-way ANOVA with Bonferroni multiple comparison with a post hoc test (multiple-group comparison). Graphs were prepared and *p*-values were calculated using GraphPad Prism 9.4.1 (San Diego, CA, USA).

## 3. Results

### 3.1. Conjugation and Quality Control of Immunoconjugate

The conjugation reactions to yield DFO-matuzumab and nimotuzumab-PEG_6_-DMI were carried out using 2 M sodium carbonate (pH 9) and 2 M HEPES (pH 8) buffers, respectively, and they resulted in clear solutions with no particulate matter/milky appearance. The purity of both immunoconjugates was confirmed on HPLC which showed similar profiles for the conjugated and unconjugated antibodies with a purity > 96% (Appendix A). This was further confirmed using a bioanalyzer (Appendix A) with molecular weights of 148.1, 151.0 (3.8 DFO/antibody), 155.8, and 161.0 kDa (3.8 PEG_6_-DM1 drug molecules/antibody) for matuzumab, DFO-matuzumab, nimotuzumab, and nimotuzumab-PEG_6_-DM1, respectively.

The effect of conjugation on the binding of these antibodies was studied in EGFR-positive DLD-1 and SNU-C2B cells using flow cytometry. The fluorescent intensity showed that there were no significant changes in the binding affinity of the conjugated antibodies compared with that of their unconjugated counterparts (Figure 1A,B and Appendix A). Mean fluorescent intensity (MFI) was converted to percentage bound and plotted against the concentration of the immunoconjugates to calculate the K_D_ and EC_50_ values in both cell lines (Figure 1C,D and Appendix A). The estimated K_D_ values were 5.7 ± 1, 11.4 ± 3, 5.9 ± 1, and 6.2 ± 2 nM for nimotuzumab, nimotuzumab-PEG_6_-DM1, matuzumab, and DFO-matuzumab, respectively. The estimated EC_50_ values were 5.9 ± 1, 11.7 ± 3, 5.2 ± 2, and 3.2 ± 3, nM for nimotuzumab, nimotuzumab-PEG_6_-DM1, matuzumab, and DFO-matuzumab, respectively. There were no significant differences in K_D_ and EC_50_ between nimotuzumab and nimotuzumab-PEG_6_-DM1 (*p* = 0.06). Additionally, there was no significant difference between matuzumab and matuzumab-DFO (*p* = 0.07).

### 3.2. Radiolabeling and Radioligand Binding Assay

The radiochemical yield of ^89^Zr-matuzumab was ≥80% at a specific activity of 0.5 MBq/µg. A radiochemical purity of ≥95% was obtained for ^89^Zr-matuzumab after purification as confirmed using iTLC and SEC radioHPLC. Binding of ^89^Zr-matuzumab to EGFR-positive DLD-1 cell line was studied using a radioligand binding assay. The estimated K_D_ and B_max_ were measured by plotting CPM counts against concentration (Figure 2A,B). K_D_ and B_max_ were 3.1 ± 0.57 nM and 22,341 ± 71, respectively.

To determine the epitope specificity of matuzumab and nimotuzumab, cells were pre-blocked with nimotuzumab or matuzumab prior to incubation with ^89^Zr-matuzumab (Figure 2B). There was a significant difference in binding between ^89^Zr-matuzumab and ^89^Zr-matuzumab + matuzumab (*p* = 0.0001) but not between ^89^Zr-matuzumab and ^89^Zr-matuzumab + nimotuzumab pre-blocked (*p* > 0.999). The results indicate that matuzumab and nimotuzumab bound to different epitopes of EGFR.

### 3.3. In Vitro Cytotoxicity

The cytotoxicity of nimotuzumab and nimotuzumab-PEG_6_-DM1 was studied using live-cell imaging in cell lines with different EGFR densities: DLD-1 > HT-29 > SNU-C2B > SW620. The cells were treated with nimotuzumab or nimotuzumab-PEG_6_-DM1, and the dose-response was monitored for 72 h using an Incucyte S3 live-cell imager. The red florescent count which is a function of cell death was converted to percentage inhibited and plotted against the concentration of the antibodies (Figure 3A–D and Appendix A). The IC_50_ for nimotuzumab-PEG_6_-DM1 vs. nimotuzumab was 8.1 ± 1.8 vs. 9.2 ± 1.0 nM, 20.1 ± 1.3 vs. 101.7 ± 2 nM, 66.4 ± 4.5 vs. 211.1 ± 6.5 nM, and 362.8 ± 3 vs. 655.4 ± 3.1 nM for HT-29, SNU-C2B, DLD-1, and SW620 cell lines, respectively. Despite the low IC_50_ of HT-29, the absolute number of red counts indicated that cell death was ≥10 fold lower than in DLD-1 with similar receptor density. Except for HT-29, in all the cell lines tested, the IC_50_ values of nimotuzumab-PEG_6_-DM1 were significantly lower (*p* < 0.05) than those of nimotuzumab, confirming the enhanced cytotoxicity of the ADC.

### 3.4. MicroPET/CT Imaging, Biodistribution, and Pharmacokinetics

MicroPET/CT imaging of ^89^Zr-matuzumab showed persistently high uptake in DLD-1, HT-29, and MDA-MB-231 tumor xenografts (Figure 4A,B) at all time points. As early as 24 h, the accumulation of ^89^Zr-matuzumab was visible in all xenografts and increased over time. The tumor-to-other-organ ratios increased over time. The highest tumor uptake was observed at 120 h (14.7% and 19.1% IA/g for DLD-1 (high EGFR expression) and HT-29 (high EGFR expression), respectively). Minimal uptake was observed in the case of SW620 xenograft which had lower EGFR expression as compared to that of the other two xenografts (Figure 4C). Pre-dosing mice bearing an MDA-MB-231 xenograft using unlabeled matuzumab significantly abrogated tumor uptake (5.1 ± 1.4% IA/g vs. 12.8 ± 1.2% IA/g with pre-dosing and without pre-dosing, respectively, *p* < 0.05), indicating in vivo specificity of ^89^Zr-matuzumab (Figure 4D).

After imaging, the mice were euthanized at 120 h post-injection. Additional groups (n ≥ 3) of mice were injected with ^89^Zr-matuzumab followed by biodistribution at 24 h post-injection (Figure 5A–C). Tumor uptake was dependent on EGFR density, with the HT-29 xenograft having the highest expression with the highest tumor uptake and SW620 xenograft having the lowest: HT-29 (19.1 ± 1.4% IA/g) > DLD-1 (14.7 ± 0.3% IA/g) > MDA-MB-231 (12.8 ± 1.2% IA/g) > SW620 (4.4 ± 0.9% IA/g). In the DLD-1 (right)/HT-29 (left) model, ^89^Zr-matuzumab was cleared in almost every organ at 120 h except from the liver (10.8 ± 1.9% IA/g) and spleen (15.8 ± 3.0% IA/g). Tumor-to-muscle ratios were 16:1, 12.2:1, 12.2:1, and 4.4:1 for HT-29, DLD-1, MDA-MB-231, and SW620, respectively. Additionally, tumor-to-blood ratios at 24 h were 2:1, 1.5:1, 1.7:1, and 1:1.4 for HT-29, DLD-1, MDA-MB-231, and SW620, respectively, and this improved at 120 h and was 3.6:1, 2.8:1, 2:1, and 0.7:1 for HT-29, DLD-1, MDA-MB-231, and SW620, respectively.

To understand the epitope specificity of matuzumab and nimotuzumab, mice bearing EGFR-positive HT-29 (left flank) and DLD-1 (right flank) xenografts were pre-dosed vail a tail vein with 50-fold excess (compared with imaging dose) nimotuzumab, followed by a tail vein injection of ^89^Zr-matuzumab. The mice were imaged (up to 120 h post-injection) followed by biodistribution immediately after imaging (Figure 6A,B). Tumor uptake of ^89^Zr-matuzumab in HT-29 and DLD-1 xenografts was not affected by nimotuzumab pre-dosing, showing that the two antibodies bound to different epitopes and could be used simultaneously for imaging and therapy.

^89^Zr-matuzumab exhibited biphasic clearance with fast distribution clearance of t_½α_ of 3.8 ± 0.27 h and a slow clearance of t_½β_ of 165 ± 14 h (Figure 7, Table 1).

### 3.5. Efficacy of Nimotuzumab-PEG_6_-DM1 and Epitope-Specific Monitoring of Treatment Response Using ^89^Zr-Matuzumab

We studied the efficacy of nimotuzumab-PEG_6_-DM1 using an EGFR-positive SNU-C2B xenograft at a dose of 15 mg/kg. Nimotuzumab-PEG_6_-DM1 was administered intravenously on days 0, 6, and 11, and the efficacy was evaluated by measuring tumor volume using a caliper and microPET/CT imaging using ^89^Zr-matuzumab. The images were acquired at the start of treatment and 20 days after treatment (Figure 8A). ^89^Zr-matuzumab PET/CT could be used to measure changes in tumor volume between the groups. Rapid tumor growth was observed with mice in the saline group (average tumor size on day 51 was 851.9 ± 723 mm^3^). Two-thirds of mice treated with nimotuzumab showed some therapeutic response, evident by the slow growth (average tumor size on day 51 was 655.9 ± 270.5 mm^3^) of the tumor xenograft. Two-thirds of mice treated with nimotuzumab-PEG_6_-DM1 (average tumor size on day 51 was 40.6 ± 57.5 mm^3^) showed complete remission with no reoccurrence over the period of this study (Figure 8B,C). No treatment-related death or significant weight changes were observed in any of these groups over the period of treatment.

### 3.6. Histology and Immunohistochemistry

After day 51, the mice were sacrificed, and tumor samples were analyzed using H&E, Ki-67, and TUNEL assays. All slides were assessed by an experienced pathologist. The nimotuzumab-PEG_6_-DM1 treated group showed more tumor necrosis than the nimotuzumab and saline-treated groups did, indicating that nimotuzumab-PEG_6_-DM1 was able to cause more tumor suppression in the mice than nimotuzumab (Figure 9A). Tumors collected from nimotuzumab-PEG_6_-DM1 showed increased TUNEL staining compared with that in nimotuzumab and saline-treated groups (Figure 9A,C). A significant difference was observed between the control and nimotuzumab-PEG_6_-DM1 (*p* = 0.035) groups but not with the nimotuzumab group (*p* = 0.364). This implies that more apoptotic cell death occurred in the nimotuzumab-PEG_6_-DM1-treated tumors than in those treated with nimotuzumab and saline. In addition, we found a significant difference between Ki-67 staining of tumors from control and nimotuzumab-PEG_6_-DM1 groups (*p* = 0.0045), and between control and nimotuzumab groups (*p* = 0.0001) (Figure 9A,B).

## 4. Discussion

To the best of our knowledge, this study demonstrates for the first time the use of an anti-EGFR probe ^89^Zr-matuzumab and an anti-EGFR therapeutic agent nimotuzumab-PEG_6_-DM1 for simultaneous epitope-specific imaging and therapy of a receptor overexpressed on cancer cells. In principle, this would allow one to understand in real-time the receptor expression of primary tumor and metastatic lesions (EGFR in this case) throughout the course of the disease and target these with the therapeutic antibody conjugate using an ADC and/or radioimmunoconjugates bearing the different target epitopes of the same receptor.

Although the use of ADCs for the treatment of cancer have gained visibility over the years, their use has been limited by the fact that most of these ADCs are a substrate for MDRI [26,35,36]. DM1, a drug of choice for many ADCs, is an MDRI substrate. Kovtun et al. [26] report that improving the hydrophilicity of the drug-linker using a hydrophilic linker such as PEG can make the drug-linker in the ADC a poor substrate for MDRI [27]. In addition to becoming a poor MDR1 substrate, the higher hydrophilicity allows for the conjugation of multiple drugs to the antibody (higher drug-to-antibody ratio (DAR)) without adversely affecting the pharmacokinetics of the biology. A previous study from our group demonstrated the efficacy of nimotuzumab-PEG_6_-DM1 with a DAR of 3–4, hence the choice of this immunoconjugate for this work [30,34].

We discovered serendipitously that matuzumab bound to EGFR at an epitope different from that of nimotuzumab. To develop epitope-specific imaging and therapeutic agents, we first needed to develop the matching imaging ^89^Zr-matuzumab agent for the therapeutic immunoconjugate nimotuzumab-PEG_6_-DM1. We previously developed ^89^Zr-nimotuzumab as an anti-EGFR imaging agent which is currently in a phase I clinical trial (NCT04235114) [37]. Therefore, in the current study, our initial objective was to develop and validate ^89^Zr-matuzumab for the first time. Matuzumab was radiolabeled with ^89^Zr via a DFO chelator, followed by characterization using flow cytometry, radioligand binding assays, and HPLC. In vitro characterization of the imaging and therapeutic agent showed low nanomolar binding to EGFR-positive colorectal cancer and breast cancer cells with (DLD-1, SNU-C2B, SW620) or without (HT-29 and MDA-MB-231) mutations in KRAS. ^89^Zr-matuzumab microPET/CT imaging and biodistribution studies show that the highest uptake of the imaging agent is observed in HT-29 despite this cell line/xenograft having lower EGFR density than DLD-1. A similar result was observed by Achmad et al. [38] who showed that KRAS wild-type xenografts consistently had a higher uptake of the anti-EGFR immunoPET agent ^89^Zr-cetuximab than mutant xenografts did. HT-29 shows higher tumor accumulation of ^89^Zr-cetuximab than DLD-1 does [38]. This observation is consistent with the findings from others that in KRAS mutant cells there is consistent loss of basolateral EGFR localization [5]. Imaging and biodistribution studies show persistent uptake of ^89^Zr-matuzumab in all xenografts except SW620, which has low expression of EGFR compared with that in HT-29. In vitro competitive binding assays showed that ^89^Zr-matuzumab bound to a different epitope than nimotuzumab, confirming they can both be used biparatopically for imaging and therapy (Figure 2B). This epitope specificity was confirmed in vivo using HT-29 and DLD-1 xenografts. Tumor uptake of mice bearing both xenografts on the right and left flanks was not affected by pre-dosing with excess nimotuzumab (Figure 6A,B).

KRAS is an intracellular effector molecule that routs ligand-bound EGFR to the nucleus, where it stimulates proliferation. Mutations in the KRAS oncogene (present in >40% of mCRC) lead to constitutive overactivation of EGFR and drive de novo resistance to anti-EGFR drugs [5,6,7]. mCRC has five major types of KRAS mutations, namely KRAS^G12D^ (34.2%), KRAS^G12V^ (21%), KRAS^G13D^ (20%), and KRAS^G12C^ (8.4%) [39]. In EGFR-positive mCRC patients with wild-type KRAS, the addition of anti-EGFR antibodies (e.g., cetuximab) to chemotherapy results in small—albeit significant—improvements in survival; there is little to no observed benefit in patients with KRAS mutations [22,40,41,42]. However, not all KRAS mutations are equal, as some are a bit more sensitive than others to anti-EGFR monoclonal antibody and tyrosine kinase inhibitor treatments. In colorectal cancer patients with different KRAS mutations, KRAS^G13D^ shows some sensitivity to anti-EGFR cetuximab, albeit less than wild-type did [22,41]. Similarly, preclinical work using DLD-1 with KRAS^G13D^ shows some sensitivity to anti-EGFR antibodies. We previously showed that mice bearing a DLD-1 xenograft treated using nimotuzmab-PEG_6_-DM1 resulted in complete tumor cure in 4/6 of the animals, which was sustained for 6 months [30]. In vitro, we investigated the cytotoxicity of nimotuzumab-PEG_6_-DM1 in KRAS mutant cell lines. We showed that the ADC was potent in vitro against DLD-1 (IC_50_ 80.1 nM), as well against other nonresponsive mutants such as KRAS^G12V^ (SW620, IC_50_ 691.4 nM) and KRAS^G12D^ (SNU-C2B, IC_50_ 20.1 nM), using live cell imaging. However, in spite of the high expression of EGFR, nimotuzumab-PEG_6_-DM1 was not very cytotoxic to HT-29 cells, as evident in the relatively lower number of cell death events (red counts) recorded.

For the first time, we studied the in vivo effectiveness of nimotuzumab-PEG_6_-DM1 in a KRAS^G12D^ SNU-C2B xenograft model and evaluated the feasibility of using ^89^Zr-matuzumab PET/CT to monitor the response to treatment. Mice were divided into three groups and treated with either saline, nimotuzumab (15 mg/kg), or nimotuzumab-PEG_6_-DM1 (15 mg/Kg). The mice were given three doses on days 0, 6, and 11, and tumor volumes were measured using a caliper and microPET. We previously confirmed that this dose was safe [30]. Mice were imaged before and 20 days after treatment to evaluate the response. Two-thirds of SNU-C2B tumor-bearing mice treated with nimotuzumab-PEG_6_-DM1 were completely cured with no tumor regrowth by day 51. Naked nimotuzumab had a minimal effect on tumor growth but no complete cure. MicroPET images after 20 days of treatment showed a decrease in tumor volume as well.

Histological evaluation showed a variable extent of necrosis in the tumors, mostly being zonal/geographic in pattern. While the tumor population had monomorphous appearance in all animals, the tumor areas surrounding necrosis showed greater polymorphism, including cells with anaplastic morphology. Tumors treated using nimotuzumab-PEG_6_-DM1 showed a greater extent of necrotic and apoptotic cell death, and reduced proliferation which is characteristic of maytansine-derived ADCs.

## 5. Conclusions

We described for the first time the use of biparatopic antibodies to image and treat mice bearing EGFR-positive tumors. This was possible because of the epitope-specific binding of ^89^Zr-matuzumab and the effectiveness of nimotuzumab-PEG_6_-DM1 ADC against EGFR-positive cells. Despite being previously described as undruggable due to their lack of sensitivity to anti-EGFR inhibitors such as antibodies and TKIs, we showed that ^89^Zr-matuzumab had very high tumor uptake in KRAS mutant xenografts despite the limited basolateral availability of EGFR due to KRAS mutations. This high uptake resulted in significant in vitro growth inhibition and tumor cure in vivo in a KRAS^G12D^ SNU-C2B xenograft using the ADC. Future work will study the effectiveness of the ADC in low EGFR-expressing KRAS^G12V^ SW620 and patient-derived xenograft models.

## Figures and Tables

**Figure 1 pharmaceutics-14-01917-f001:**
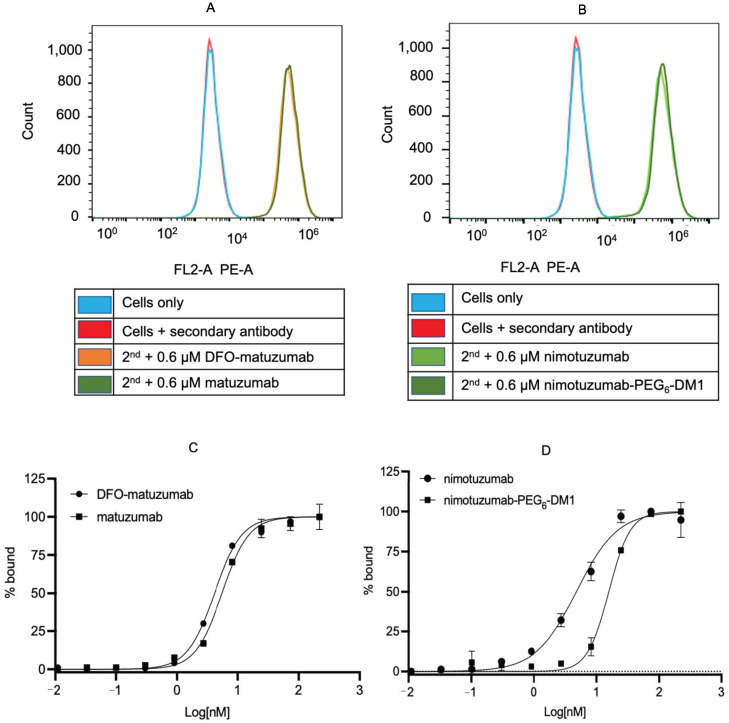
Flow cytometry of immunoconjugates in DLD-1 cells. (**A**) Nimotuzumab and nimotuzumab-PEG_6_-DM1, and (**B**) matuzumab and matuzumab-DFO at 0.6 µM. Cells with the secondary antibody and untreated cells were used as control. (**C**,**D**) A 10-point saturation binding that allowed for the determination of K_D_ and EC_50_ of the immunoconjugates. The mean fluorescent intensity was converted to percentage bound and plotted against concentration, and a nonlinear curve fitting was used to estimate K_D_ and EC_50_. All experiments were carried out in duplicates.

**Figure 2 pharmaceutics-14-01917-f002:**
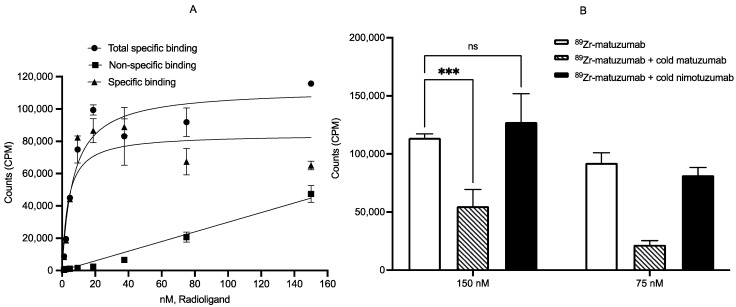
Radioligand binding assay of ^89^Zr-matuzumab in DLD-1 cells. (**A**) Total, nonspecific, and specific binding of ^89^Zr-matuzumab in DLD-1 cells. The specific binding was gotten from the subtraction of nonspecific binding from total binding. (**B**) Epitope-specific competitive binding of nimotuzumab and matuzumab to EGFR. Binding of ^89^Zr-matuzumab to EGFR was not blocked by excess of nimotuzumab indicating epitope specificity (ns = not significant (*p* > 0.05), *** = highly significant (*p* <<< 0.05)).

**Figure 3 pharmaceutics-14-01917-f003:**
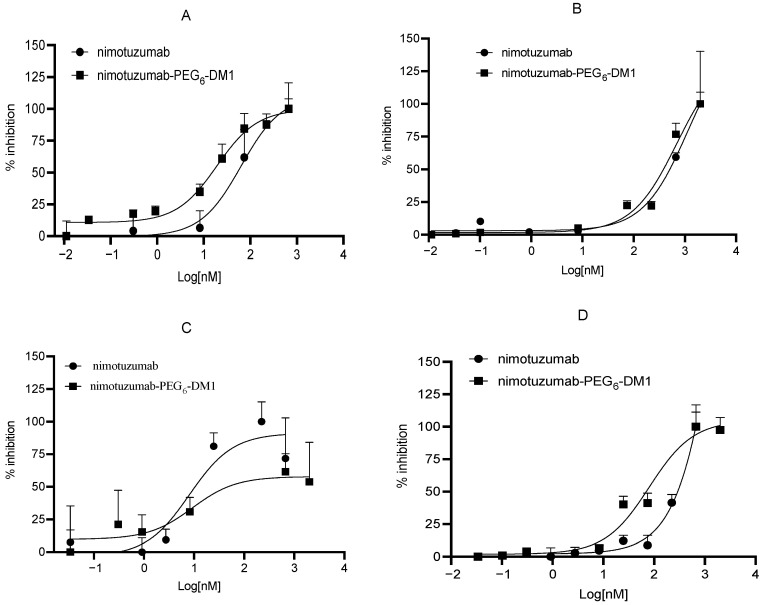
In vitro cytotoxicity of nimotuzumab and nimotuzumab-PEG_6_-DM1 against (**A**) SNU-C2B, (**B**) SW620, (**C**) HT-29, and (**D**) DLD-1 cell lines. The IC_50_ values were calculated by plotting the increase in red fluorescent intensity against concentration using Prism 9. All experiments were carried out in triplicates.

**Figure 4 pharmaceutics-14-01917-f004:**
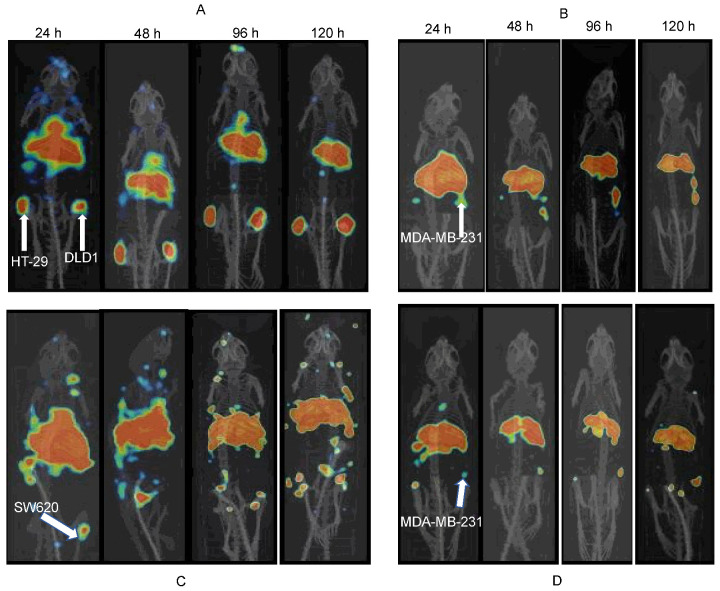
MicroPET imaging of mice bearing EGFR-positive xenografts following a tail vein injection of ^89^Zr-matuzumab. (**A**) Mouse bearing DLD-1 (right flank) and HT 29 (left flank), (**B**) MDA-MB-231, (**C**) SW620, and (**D**) MDA-MB-231 after pre-dosing with unlabeled matuzumab prior to ^89^Zr-matuzumab (blocking studies) xenografts.

**Figure 5 pharmaceutics-14-01917-f005:**
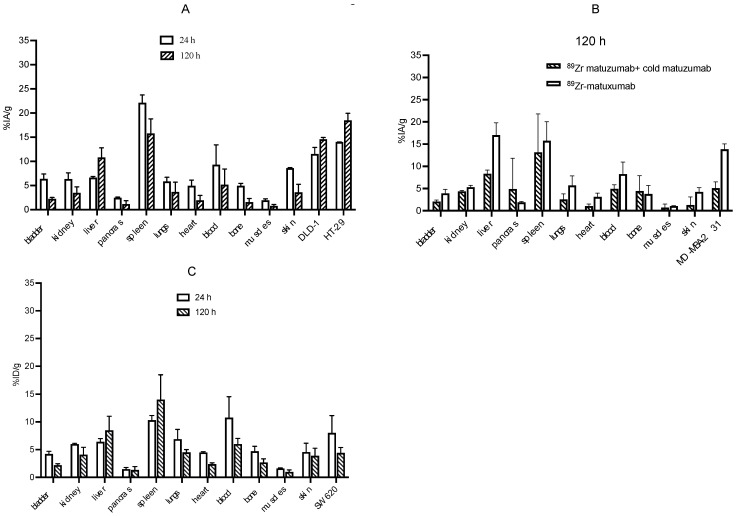
Biodistribution of ^89^Zr-matuzumab in athymic CD-1 nude mice. (**A**) Mice bearing DLD-1 (left flank)/HT-29 (right flank) xenografts at 24 and 120 h following injection of ^89^Zr-matuzumab. (**B**) Mice bearing an MDA-MB-231 xenograft were pre-dosed with a large excess (50-fold of the imaging dose) of unlabeled matuzumab 24 h prior to a tail vein injection of ^89^Zr-matuzumab. (**C**) Mice bearing SW620 xenografts at 24 and 120 h following injection of ^89^Zr-matuzumab.

**Figure 6 pharmaceutics-14-01917-f006:**
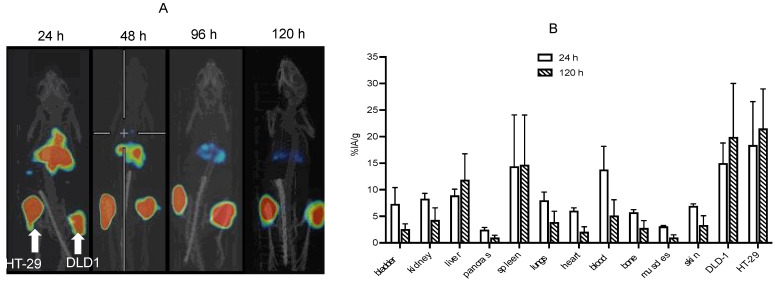
In vivo epitope-specific imaging and biodistribution. (**A**) CD-1 athymic nude mice bearing EGFR-positive DLD-1 (left flank) and HT-29 (right flank) were pre-dosed with excess (50-fold of the imaging dose) of nimotuzumab 24 h prior to administration of ^89^Zr-matuzumab followed by microPET/CT imaging. (**B**) Ex vivo biodistribution of the mice at 120 h post-injection.

**Figure 7 pharmaceutics-14-01917-f007:**
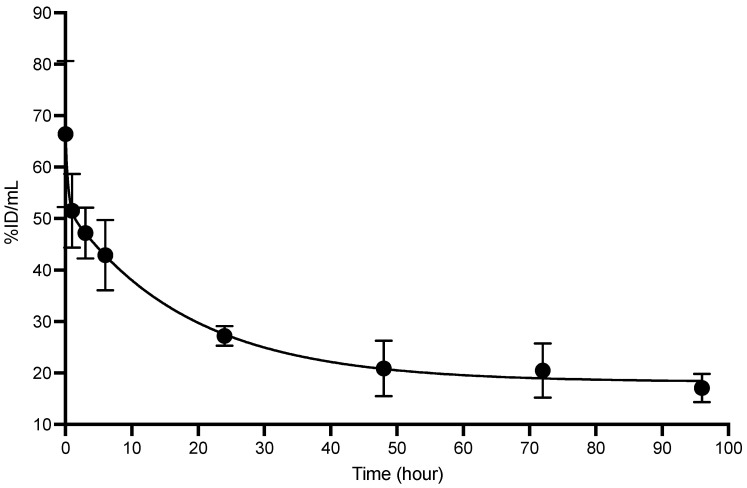
Pharmacokinetics of ^89^Zr-matuzumab in healthy Balb-C mice. Blood values are expressed in percentage of injected activity per milliliter (% IA/mL) and were used to generate pharmacokinetic parameters.

**Figure 8 pharmaceutics-14-01917-f008:**
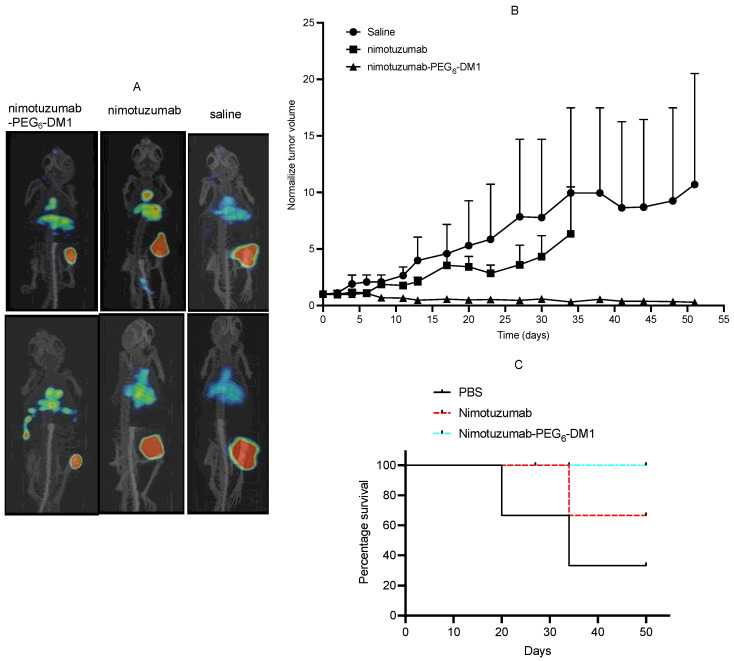
Epitope-specific treatment of mice bearing an EGFR-positive colorectal cancer xenograft SNU-C2B using nimotuzumab-PEG_6_-DM1, and monitoring of response to treatment using ^89^Zr-matuzumab. (**A**) MicroPET/CT images of ^89^Zr-matuzumab on days 0 (upper panel/row) and 20 (lower panel/row) after treatment. Mice were treated on days 0, 6, and 11 using nimotuzumab-PEG_6_-DM1 ADC. (**B**) Tumor growth curves of mice treated using nimotuzumab-PEG_6_-DM1, nimotuzumab, or untreated (saline), and (**C**) Kaplan–Meier survival curves of treated and untreated mice.

**Figure 9 pharmaceutics-14-01917-f009:**
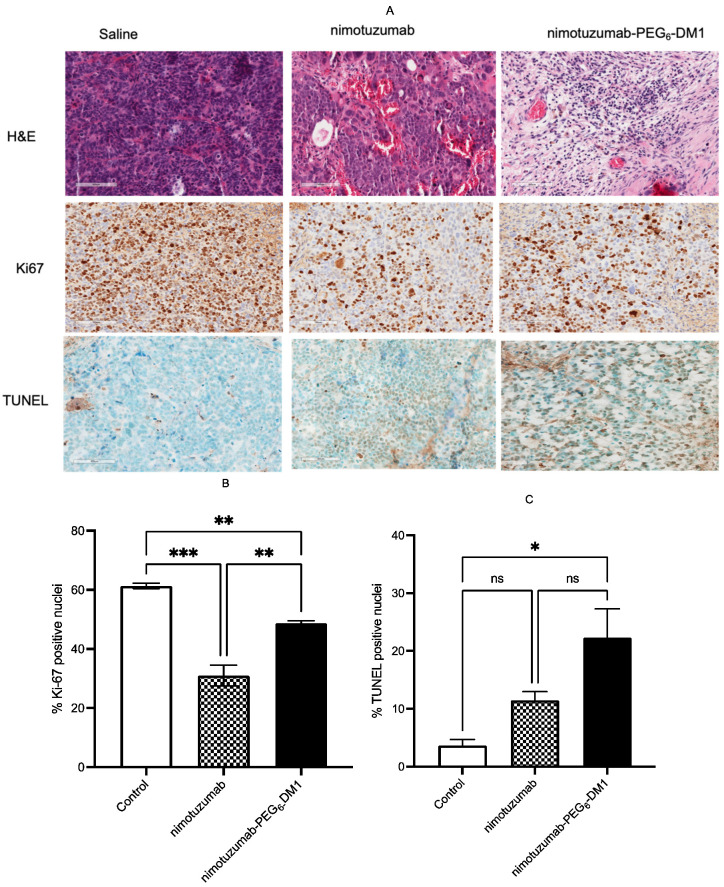
Histological analyses of in vivo treated and untreated mice. (**A**) H&E, TUNEL, and Ki-67-stained tumor sections collected from mice at 51 days after treatment Scale bar 100 µm). (**B**) Bar chart representation of Ki-67-positive nuclei, and (**C**) bar chart representation of TUNEL positively stained nuclei. (ns = not significant (*p* > 0.05), * = significant (*p* < 0.05), ** = significant (*p* << 0.05), *** = highly significant (*p* <<< 0.05).

**Table 1 pharmaceutics-14-01917-t001:** Pharmacokinetic parameters of ^89^Zr-matuzumab ± SD.

Compound	$t_{½}\alpha$	$t_{½}\beta$	AUC (% IA. h/mL)	V_1_ (mL)	CL × 10^−2^ (mL/h)
^89^Zr-matuzumab	3.8 ± 0.3	165 ± 14	5857.5 ± 20	4.1 ± 0.1	1.7 ± 0.5

## Data Availability

The data presented in this study are available on request from the corresponding author.

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
