# Peer review of "Simultaneous Imaging and Therapy Using Epitope-Specific Anti-Epidermal Growth Factor Receptor (EGFR) Antibody Conjugates"

_pharmaceutics, 2022, doi:10.3390/pharmaceutics14091917_

Round 1

Author Response

In the manuscript “Simultaneous imaging and therapy using epitope-specific anti-2 epidermal growth factor receptor (EGFR) antibody conjugates”, the authors present the results of a study aimed at using the EGFR receptor as a target for ADC-immunotherapy and imaging, by exploiting two already existing antibodies (nimotuzumab and matuzumab) capable of binding different epitopes. The experimental model used for in vitro and in vivo studies is colorectal carcinoma cell lines expressing either wild type or mutated KRAS.

The relevance of the presented results could benefit from a more accurate presentation and statistical analysis.

Below, a point-by-point list of observations.

  1. 6, line 260: The estimated KD and Bmax were measured by plotting the CPM counts against concentration (Fig. 2A and 2B). Panel B represents Epitope-specific competitive binding results.

Response: This has been corrected

  1. 6 line 270: The estimated KD and Bmax KD = 3.1 nM; Bmax = 22341. Are the values average of independent experiments? If this is the case, please report SDs. Moreover, looking at figure 2A, the estimated Bmax and KD values seem to poorly correlate with the experimental data for specific binding. For example, significative specific binding seems to be detected above 40 nM radioligand. Please, clarify.

Response: SDs have been added. The KD value was calculated using graphpad prism

  1. 6 line 273. Figure 2C. Panel C for figure 2 is not included in the manuscript.

Response: This has been corrected. There is no Figure 2C

  1. 6 line 273. There was significant difference between 89Zr-matuzumab and 89Zr-matuzumab + matuzumab (p = 0.0001) but not for 89Zr-matuzumab + nimotuzumab pre-blocked (p > 0.999). It is not clear to me the parameter and the values object of the comparison.

Response: The object of comparison is the radioligand binding of 89Zr-matuzumab to EGFR positive DLD-1 cells in the presence of nimotuzumab or matuzumab to establish the epitope specificity of these two antibodies. This comparison is with regards to absolute binding in the presence or not of the competitor.

  1. 7, line 294. The IC50 for nimotuzumab-PEG6-DM1 vs nimotuzumab was 20.1 +/- 1.3, vs 101.7 +/- 2 nM, 66.4 +/- 4.5 vs 211.1 +/- 6.5 nM, and 362.8 +/- 3 vs 655.4 +/- 3.1 nM for SNU-C2B, DLD-1, and SW620 cell lines, respectively. The IC50 for nimotuzumab-PEG6-DM1 vs nimotuzumab for HT-29 cell line is not reported, even though cytotoxicity of nimotuzumab-PEG6-DM1 and nimotuzumab was tested also in this cell line (line 289: The cytotoxicity of nimotuzumab and nimotuzumab-PEG6-DM1 was studied using live-cell imaging in cell lines with different EGFR densities: DLD-1 > HT-29 > SNU-C2B > 290 SW620., and figure 3 C anD figure S4).

Response: HT-29 cells displayed very low cytotoxic activity for nimotuzumab and nimotuzumab-PEG6-DM1 and hence IC50 value could not be estimated

  1. 7, line 297. In all the cell lines tested, the IC50 values of nimotuzumab-PEG6-DM1 was significantly lower than that of nimotuzumab confirming the enhanced cyto-298 toxicity of the ADC. Please, report the p values for each cell line, comprising also HT-29.

Response: P value has been added. Despite having high EGFR expression HT29 did not respond well to nimotuzumab-PEG6-DM1 and nimotuzumab as evident by the relatively low number of cell death events. This cell line is notoriously aggressive as previously reported by Hartimath et al. Oncotarget: 10: 1031-1044

  1. 8, line 316. Pre-dosing mice bearing MDA-MB-231 xenograft using unlabeled matuzumab significantly abrogated tumor uptake (5.1% IA/g vs 12.8% IA/g for with pre-dosing and without pre-dosing, respectively) indicating in vivo specificity of 89Zr- matuzumab (Fig. 4D). SDs and p-values are not indicated. Please, add to the text.

Response: This has been corrected

  1. 8, line 322. (Fig. 5A - D) Panel D is not in the present manuscript.

Response: This has been corrected

  1. 8, line 323. with the highest tumor uptake and SW620 xenograft having the least: Least” should be lower?:

Response: This has been corrected

  1. 8, line 325. 89Zr-matuzumab was cleared in almost every organ at 120 h except from the liver (10.8 % IA/g) and spleen (15.8 % IA/g). At which model these values are referred, DLD-1? SDs and statical analysis are missing in the text, significance could be represented also in figure 5 A-C to make the analysis of the data easier for the reader.

Response: SDs and the model used have been added.

  1. 10, line 349. xenografts were pre-dosed vail a tail vein with 100-fold excess (compared with imaging dose) nimotuzumab In figure 6 caption (pg 11, line 361) the pre-dose excess is 50-fold, as well as reported in previous experiments. Please, clarify.

Response: This was 50-fold. This has been corrected

  1. 11, line 366 t½α of 3.8 +/- 0.27 h and a slow clearance of t½β of 165 +/- 14 h (Fig. 6, Table 1) In the present manuscript the pharmacokinetics curve is represented in Figure 7.

Response: This has been corrected

  1. Table 1. 89Zr-matuzumab 3.8 +/ 0.3 Maybe it should be +/-, not +/?.

Response: This has been corrected

  1. 12 paragraph 3.5. Comparison of PET/CT imaging with caliper measurements are not reported. How 89Zr-matuzumab use as monitoring agent has been validated? In the paragraph, SDs and statistics are not reported.

Response: Because the imaging study was only done at two timepoints (immediately before treatment and 20 days after), we do not feel this was enough data points to make a meaningful comparison between PET/molecular imaging and caliper measurement. This experiment was meant to be feasibility study to show intense/non-competitive uptake of the tracer during treatment using the ADC. As such we made the decision not to quantify the images. In a larger future study, we would be able to make a meaning quantitative comparison.

  1. 13, line 408. A significant difference was observed between the control and nimotuzumab-PEG6-DM1 (p = 0.035) groups but not with the nimotuzumab group (p = 0.364). This implies more apoptotic cell death occurred in nimotuzumab-PEG6-DM1 treated tumor than those treated with nimotuzumab and saline. The sentences are contradictive, please amend as apoptosis is not significantly different between nude and ADC nimotuzumab.

Response: nimotuzumab-PEG6-DM1 lead to cell death that was significantly more than control/non-treated but was not significantly more than nimotuzumab. Nimotuzumab is potent leading to cell death as quantified in the bar chats. We do not think this is contradictory, as it is supported by in vitro and in vivo data

  1. 14, Discussion. The discussion paragraph can benefit from a revision with the aim of make it more synthetic and with a more to-the-point discussion of the results.

Response: We believe that the relevance of the results has been well discussed

  1. In general, the authors should go through the entire manuscript and check for text consistency, proper and consistent use of italic and appendix/subscript, for typos and for completeness and clarity of sentences.

Response: This has now been well addressed throughout the paper

Reviewer 2 Report

The manuscript titled "Simultaneous imaging and therapy using epitope-specific anti-epidermal growth factor receptor (EGFR) antibody conjugates" submitted to Pharmaceutics (pharmaceutics-1866622), describes a very interesting approach related to extending study refered to Matuzumab and nimotuzumab. Authors developed Zr-matuzumab as a PET probe for diagnosis/monitoring of response to treatment of a non-competing anti-EGFR nimotuzumab antibody drug conjugate (ADC) using mouse colorectal cancer (CRC) xenografts. Authors also developed Zr-matuzumab and performed quality control in EGFR-positive DLD-1 cells. The topic is very actual, original and interesting. Abstract and introduction is well written, Methods and Materials are very well organize. Results and Discussion are presented very clearly and describe all experiments very well. Tables and figures are presented correctly. It looks that author very carefully designed experiments and that they have a wide knowledge about methodology and instruments used in the experiments.. 

Authors concluded that Matuzumab and nimotuzumab are non-competing and can be used simultaneously.

Author Response

We thank the reviewer for his/her efforts in reviewer the paper. There were no comments to respond to.

Reviewer 3 Report

The article fits the scope of the journal very well and is of high interest to readers. This study contributes to the knowledge about molecular design of targeting agents for radionuclide imaging and therapy in EGFR-expressing cancers. An interesting and well-executed study. The manuscript is well written. I believe that the manuscript could be accepted for publication after Minor revisions.

1.       Check for typos and spelling in Materials and Methods section

2.       Section 2.4. Please include a detailed protocol of DFO conjugation and radiolabeling. Please add information and radiochemical yield and purity to Results. This would be very helpful for other researchers working with radiolabeling of mAbs.

3.       Section 2.5. Establishment of xenografts, specify how many cells was implanted for each tumor model.

4.       Page 12 line 382-383. You state that “89Zr-matuzumab 382 PET/CT can accurately measure changes in tumor volume between the groups.”, however, no data was provided to support this statement. Could you please add the data (quantification from PET imaging?) or update the statement?

5.       Supplementary information. Figure S1: increase the size of all graphs and font size, not readable.

Author Response

The article fits the scope of the journal very well and is of high interest to readers. This study contributes to the knowledge about molecular design of targeting agents for radionuclide imaging and therapy in EGFR-expressing cancers. An interesting and well-executed study. The manuscript is well written. I believe that the manuscript could be accepted for publication after Minor revisions.

  1. Check for typos and spelling in Materials and Methods section

Response: These have been updated

  1. Section 2.4. Please include a detailed protocol of DFO conjugation and radiolabeling. Please add information and radiochemical yield and purity to Results. This would be very helpful for other researchers working with radiolabeling of mAbs.

Response: The protocol has been added

  1. Section 2.5. Establishment of xenografts, specify how many cells was implanted for each tumor model.

Response: Corrected

  1. Page 12 line 382-383. You state that “89Zr-matuzumab 382 PET/CT can accurately measure changes in tumor volume between the groups.”, however, no data was provided to support this statement. Could you please add the data (quantification from PET imaging?) or update the statement?

Response: The statement has been updated

  1. Supplementary information. Figure S1: increase the size of all graphs and font size, not readable. Response: Corrected